# Assessment of Indoor Air Quality and Users Perception of a Renovated Office Building in Manchester

**DOI:** 10.3390/ijerph17061972

**Published:** 2020-03-17

**Authors:** Haya Fahad Alomirah, Haruna Musa Moda

**Affiliations:** 1College of Health Sciences, Shawaik, P.O. Box 1983, Safat 13020, Kuwait; hf.alomirah@paaet.edu.kw; 2Department of Health Professions, Manchester Metropolitan University, Manchester M15 6BG, UK

**Keywords:** sick building syndrome, indoor air quality, post occupancy evaluation, questionnaire survey, instrumentation

## Abstract

Building renovations can adversely affect building occupants through the release of biological contaminants, gases and particulates. In this study, the research aim was to monitor the air quality of a renovated building and assess the impact of sick building syndrome (SBS) on the occupants. Post occupancy monitoring of the building was carried out after two months occupancy for the following environmental parameters: airborne microflora using an air sampler (SAS super 180) and a hand-held monitoring device (Graywolf advance sense IQ-610) to measure total volatile organic compounds (TVOC), CO_2_, CO and temperature and relative humidity in each office environment. In addition, an online (Qualtrics) structured questionnaire was used to assess occupants’ perceptions of the indoor environment. Results of the airborne flora showed 833 cfu/m^3^ recovered on a Malt Extract Agar (MEA) plate in the morning and 1213 cfu/m^3^ in the afternoon. A similar result was noticed on a Plate Count Agar (PCA) plate during the morning period (731 cfu/m^3^) and afternoon (1358 cfu/m^3^). Results of TVOC monitored over one week showed that the first two days of monitoring had a high reading that peaked at 10,837 ppb and that the CO_2_ concentration during that period was 1163 ppm. Online questionnaire analysis indicates that a majority of the staff who took part in the survey experienced some form of health abnormality, including headache, shortness of breath, itchy eyes/ears, loss of concentration and so on, especially in the first few weeks of returning to the office. The results from the study indicate that a large proportion (41%) of the respondents experienced thermal discomfort as a result of varying room temperature during their working hours. A high number of female participants experienced some form of SBS as compared to their male counterparts. The study findings show a direct relationship between high airborne mold counts, TVOC and adverse staff health perception of the building. The study raised a number of opportunities for estate managers to improve building performance based on occupants’ preferences.

## 1. Introduction

Indoor air quality is becoming a matter of concern to the both medics and other experts in built environments. We spend over 70% or our time inside buildings without being aware of the indoor conditions that may lead to adverse health conditions. Experts have concluded that exposure to air pollutants present in indoor air, poor air exchange with the outside environment and inadequate lighting in the building in which people work can negatively impact occupants’ health, leading to a form of health problem termed sick building syndrome (SBS). SBS is an ill-health condition experienced when people in a work environment exhibit a range of non-specific, building-related symptoms. Research related to indoor environmental conditions and how they affect staff performance, health or satisfaction has been carried out over the past decade, especially in developed countries [1,2,3], with results revealing the impact of built environments on workers’ performances and health. The target of conventional building maintenance managers is to provide a conducive working environment capable of satisfying the needs of most users. However, not all occupants may be satisfied with the state of an indoor environment due to the contributions of several factors. These factors may include gender, age, psychosocial factors or past exposure to chemical, microbial or physical agents, or the fact that various individuals react differently to the same indoor environment [2]. Poorly maintained indoor environments can increase time lost to illness and stress, and can reduce comfort and staff work pace, leading to a reduction in work output [4,5]. 

There are limited study around the role of gender and individual response to indoor environment air quality. However, from these few work done, the female gender has been reported to be an important factor in several SBS studies [6]. Recent studies by Kim et al. [2] and Karjalainen [7] have reported females as being more likely than males to report dissatisfaction of an indoor air environment and other sick building symptoms (SBS) such as fatigue, headache, irritated or dry eyes/nose/throat or skin symptoms. It has further been suggested that the reason why women report more symptoms related to SBS than men is that they tend to work in less favorable physical and psychosocial conditions than men [8]. However, it has been pointed out that the risk of overestimating gender differences without taking into account other factors (e.g., nutritional differences, fitness, ethnicity and social class) may lead to a misrepresentation of SBS impact based on gender differentiation [2].

Despite efforts that have been made to improve air quality and ventilation standards in buildings, occupants’ satisfaction level also need further investigation, as ventilation standards do not necessarily ensure a reasonable level of occupant satisfaction [5,9]. Considering that SBS symptoms are multifactorial in origin, psychosocial work characteristics such as workload and job satisfaction, and worry and reorganization, could be contributory factors that significantly impact the development of SBS among workers, as has been discussed previously by Erickson et al. [10]. Hence, an adverse psychosocial environment has the potential to make an individual give more attention to discomfort, health and other potential likely adverse sources in the physical environment. 

Vibration and noise are other factors that are likely to impact the user of an indoor environment. As stated by Frontczak and Wargocki [4], the quality of any sound in an indoor environment can be linked to several physical parameters that include the physical properties of sound and its acoustics, including sound insulation, absorption of surfaces and reverberation time. In addition, absence of or insufficient lighting in buildings has been associated with SBS. The visual comfort of building users can be influenced by a combination of luminance distribution, illuminance and its uniformity, glare, color of light, color rendering, flicker rate and amount of daylight [4,11]. It is worth noting that urban environments comprise high building densities, of which sick building syndrome (SBS) symptoms and levels of thermal comfort are important concerns from both building operation and occupant health/productivity points of view [12]. To ensure that occupant health is not adversely affected in indoor environments, investigation into these complaints is considered in this paper.

The aim of this study is to investigate the impact of workplace indoor environments as they affect occupants’ comfort, satisfaction and performances in a newly refurbished office block. In addition, this study considers gender sensitivity to prevailing environmental factors in the indoor workplace and its association with symptoms of SBS based on a case study approach using physical indoor air quality measurements, available building information, an occupant questionnaire survey and a building survey. 

## 2. Materials and Method

### 2.1. Online Questionnaire Survey

An online questionnaire survey was carried out using Qaultrics online survey software (Qaultrics Lab Inc. Seattle, WA, USA). Questions considered in the survey were centered around the respondents’ sensory perceptions of indoor environments, their control of workstation environments and their psychosocial responses to factors that impact their productivity at work. The choice of online surveying was preferred over the self-administered questionnaire, as it offered the respondent the opportunity to access the survey via a web link that was mobile technology friendly. In addition, it allowed easy access to targeted groups considered in the study design, which in this case were staff members of the same institute who were working in the same building and using a group email account that had already been created. 

### 2.2. The Study Area

The indoor air quality survey was carried out in a three-office accommodation with an average of five staff per office space. The first office sample was 72.5 m^3^ with a window area of 6.2 m^2^. The second office had a space of 111.8 m^3^ and a window area of 7.8 m^2^. The third office was 114.2 m^3^ in size, with a 8.2 m^2^ window area. The floor of each room was covered in carpet, and office furniture included chairs, desks, metal office cupboards and metal desk height pedestals. Present in each room were desktop monitors, a fixed telephone receiver and a docking station for laptops. Each workstation had a white background with anti-glare properties. LED fluorescent tubes were used to light each office room, with automatic sensors to limit energy use when not occupied. Corridors were laid with carpet, and printers were stationed in each corridor. 

### 2.3. Airborne Micro Flora Sampling Protocol

An air sampler (SAS super 180 Cherwell Laboratories, Bicester, UK) was used for air sampling. Air was aspirated at a fixed rate of 180 l/min through a sterilized perforated metal cover onto the surface of a 50-mm contact plate containing a selected agar. Malt extract agar (MEA) and plate count agar (PCA) were purchased from Oxiod, (Basingstoke, UK), and were used for air sampling in the building. Aspirated plates from the air sampler were incubated at 25 °C for 4 days or until visible growth appeared on the plates, afte which the colonies were counted using plate counter SC6 plus (Stuart Scientific, Staffordshire, UK) and expressed as colony-forming units per cubic meter. Colonies were later corrected for the statistical possibility of multiple particles passing through the same hole, according to the manufacturers’ guidance. Probable count (Pr) was used to calculate the CFU per cubic meter of sampled air (CFU/m^3^) using the equation
(1)$$x=\mathrm{Pr}x\;\frac{1000}{V}$$
where: V is volume of the air sampled, r is the colony forming units on plates, Pr is the probable count obtained by the positive hole correction and X is the colony forming units per cubic meter of air.

Isolates were sub-cultured onto MEA and potato dextrose agar (PDA) with 7.5% NaCl (v/v) to serve as an inhibitor to fast growing “spreader” mold (i.e., *Mucor* and *Rhizophus*) for subsequent identification. To identify the isolates, a block of MEA (2 × 2 cm) was inoculated with each mold spore and sandwiched between two sterile slide cover glasses, then placed in a sterile Petri dish containing cotton wool soaked with 4 mL deionized water. The culture assembly was placed in a 23 °C incubator for 4 days and thereafter disassembled, mounted on a sterile microscope slide, stained with lactic blue and observed under a light microscope. The morphological appearance of the mold isolates were compared with standard reference texts that were based on growth and colony characteristics on media plates and microscopic examinations as described by Samson et al. [13].

### 2.4. Indoor Air Quality Monitoring 

Office accommodation housing staff were selected on the ground and second floor for the survey. Real time monitoring of the indoor environment (temperature and relative humidity, total volatile organic compounds (TVOC), CO_2_ and CO) was performed using GrayWolf Advanced Sense IAQ, Indoor Air Quality Instrument (IQ-610) GrayWolf Sensing Solutions, Shelton, USA. As described in our earlier work [5], to guard against sensitivity loss from either sensor poisons or suppressors present in the indoor environment monitored, the instrument was factory calibrated before the study. In addition, a functional (bump) test was conducted prior to the start of each sampling at each location in order to ensure that the response of the sensors was within an acceptable tolerance range of the actual concentration. Prior to the start of each sampling, the equipment was programmed and allowed to warm up for 20 min to allow all sensors to stabilize and ensure accurate reading in the location. In each sampled office, the data logger was placed in the carrier box at a height of 1 meter above the floor and away from any heating source. This allowed the sampling head to be exposed to the indoor air whilst protecting it from tampering.

In each room monitored, occupants were advised to continue with their normal indoor activities while sampling took place (seven days). The instrument was programmed to record the selected environmental parameters at 10-min time intervals for the duration of the survey. The average office occupancy was five staff per each sampled location. Staff were advised to continue with their normal work schedules when in the office. This was encouraged in order understand how the building responded to the behavioral patterns of the occupants. 

A dosimeter calibrated using Quest Sound Calibrator (Quest electronics M28; Oconomowon, Wisnconsin, USA) was used to monitor the noise levels in the office environment housing 10 staff members. Microsoft Excel 2007 was used to calculate the standard deviation, analysis of variance and chi square tests of the data. 

### 2.5. Ethical Approval

Ethical approval was obtained from Manchester Metropolitan University to approach the study participants. Participants were reassured of their confidentiality and their right to withdraw from the study at any time they deemed. 

## 3. Results

Of the 106 staff who attempted to fill out the online survey, 94.3% (100) were completed. Of the respondents, 69% were female and 35% said they combined teaching and research (Table 1). 

Of participants, 76% indicated that their office environment was ventilated by means of opening a window, and a combined 53% of respondents said that their office environment was either crowded or overcrowded (Table 1). To understand the past health history of staff, the questionnaire asked who had experienced asthma prior to starting work in the organization, with 19% responding to having had experienced asthma previously. SBS among the staff is reported in Table 2 below. More than half of the respondents indicated having suffered one form of SBS in the previous three months during their period of work in the office environment. Headache was the most common form of SBS reported by the staff. Fatigue (46%) and difficulty in concentrating to execute tasks (28%) were the next most prevalent concerns raised by staff, with all responses strongly linked to work environment (Table 2). 

Responses to questions around the staff office occupation ratio revealed that more than half (53.8%) believed their office to be crowded or overcrowded. Of this number, 66.6% classifying the office space as crowded or overcrowded identified themselves as females. The environmental impacts considered in this study are presented in Table 3. Office occupancy ratio correlates to the number of staff (53.2%) who reported noise as the major concern during working hours (R = 0.55, *p* < 0.05). Noise level measured in office 1, which houses eight academic staff, gave 59 dba as the average noise level over a 90-h monitoring period, with the highest noise peak being 134 dba. In addition, temperature variation was a source of concern with 41.1% of the respondents. Average occupancy in the office spaces was at least five staff, depending on the workstation office. Visual assessment of the office space showed that most staff seated closer to windows tended to keep the windows closed, to prevent external pollutants and possible drafts from infiltrating the building. 

Table 4 shows the average measurement of environmental variables over an 8-h work period from three selected office spaces. Each office space was occupied by five or more staff and was of equal volumetric size. Data from office 1 had an average temperature of 25.8 °C, with an average CO_2_ concentration of 630 ppm. Results of airborne mold samples showed that office 1 spore counts were higher (833 cfu/m^3^) in relation to the remaining two office spaces considered in the study (Table 4). Mold genera isolated during the study included *Apergillus*, *Penicillium*, *Cladosporium*, *Ulocladium Alternaria*, *Mucor*, *Rhizophus* and yeast. 

## 4. Discussion

Based on the analysis conducted, it has been demonstrated that there were different satisfaction levels among occupants. SBS is a subject that has eluded many building managers/employers. A report from the office of national statistics [14] revealed that over 131 million days each year are lost due to sickness absence in the UK, to which elements of SBS have been attributed. It is possible that some of these absences could be avoided if employers were more familiar with SBS and its potentially adverse health effects. Although, the survey from the present study did not consider staff response as it affected sickness absence from work over the last three months, an indication of the prevalence of SBS that may contribute to ill health at work revealed that fatigue, frequent headaches, noise and varying room temperature (among others) were the recurring issues. These issues have a direct impact on the daily productivity of staff. Other key findings from the study were related to gender and SBS reporting. The survey results indicate that females who took part in the study tended to experience greater amounts of SBS compared to their male counterparts. Noise, varying room temperature, dry air and dust were the major forms of SBS concerns highlighted by females. The trend observed from the results has not been verified further to determine if it is related to individual lifestyle issues or physiological differences. Earlier studies around indoor air quality and SBS revealed that there tend to be direct relationships between these variables and gender. These studies reported similar findings where the female respondents reported less satisfaction around each variable considered [15,16]. Both physiological and metabolic rate differences have previously been reported as factors that contribute to different levels of SBS reported between genders [17,18]. 

Thermal comfort plays a significant role in the productivity of occupants of any indoor environment, and high degrees of thermal discomfort can result in loss of productivity [19]. The results from this study indicate that a large proportion (41%) of respondents experienced thermal discomfort due of varying room temperatures during their working hours. Earlier studies around the relationship between office task, temperature and productivity have indicated different optimum temperatures, with a thermal comfort impact on individual work productivity [19,20,21]. In practice, it is important that due consideration be paid to seating arrangements in relation to individual needs. 

Total volatile organic compounds (TVOCs) present in indoor environments present irritant and odorant properties, and this contributed to the response patterns observed in the study. Considering that the office spaces monitored were newly renovated, it can be seen how the amounts measured in each space (considering that paints, carpets and furniture) were newly introduced into each office space. In a given indoor environment, a concentration of TVOCs could vary and depend on the presence or absence of emission sources, and exposure to TVOCs can result in both acute and chronic health effects. Earlier studies have affirmed that pollutant loads in indoor environments are associated with a number of factors that include intensity of air exchange, emissions from various indoor and outdoor sources, temperature and humidity, ventilation systems, atmospheric air quality in the vicinity, external emission sources and so on [22,23,24]. At high concentrations, many of these TVOCs are potent narcotics and can depress the central nervous system [5,25], and exposure to these compounds can lead to irritation of the eyes and respiratory tracts, in addition to causing sensitization reactions involving the eyes, skin and lungs. Emphasis around reduction and possible elimination of emission sources should be actively encouraged in order to reduce their build up in indoor environments. 

Analysis of the mold recovered from the office spaces indicates the presence of *Apergillus*, *Penicillium* genera alongside other species. The external environment was the chief source of these molds found in the indoor air, and seasonal variations and climatic conditions contribute to increases in the number but also the types of fungal spores present in the air [26,27]. Spores of mold pathogens such as *Penicillium* and *Aspergillus* are numerous in indoor air and pose a hazard, especially among individuals with underlying health challenges. Other studies [24,25,26,28,29,30] have previously established a strong association between mold presence in buildings and reported respiratory symptoms among occupants. In addition, microbial volatile organic compounds (MVOCs) associated with mold have being a subject of concern in indoor environments [31,32,33]. One of the possible effects posed by MVOCs in indoor environments is their ability to evoke the sensory irritation of sensitive organs in hosts. Nevertheless, due to the presence of other sources of pollutants in indoor environments, little attention has been placed on their contribution to the state of indoor air quality (IAQ). Based on this, there is a need for estate managers to ensure that their proliferation in indoors environment is controlled efficiently. 

## 5. Conclusions

This study raised a number of issues that should be taken into account in order to consider the needs of different users of indoor environments. While some of the issues raised herein might be difficult to implement, especially in open-plan offices, there is a range of reasonable adjustments that could be considered in order to minimize the risk of SBS among workers and help drive productivity among employees. The study findings support the view that a multifactorial approach is needed in the prevention of SBS through the promotion of applicable physical and psychosocial measures. In addition, based on the number of issues associated with noise, temperature, presence of pollutant sources and so on, estate managers should consider the frequent monitoring and assessment of office spaces in order to help improve building performances based on occupants’ preferences. 

## Figures and Tables

**Table 1 ijerph-17-01972-t001:** Characteristics of the survey participants.

Variables	Percentage	
Age:		
18–25	3	Mean: 3 Variance: 0.47 SD:0.69
26–34	13	
35–54	66	
55–64	17	
>65	1	
Gender:		
Male	31	Mean: 1.69 Variance: 0.22 SD: 0.47
Female	69	
Occupation:		
Education-Research	10	
Education-Teaching	31	
Administration	29	
Education Teaching and Research	30	
Ventilation type:		
Mechanical	5	
Natural	76	
Natural and Mechanical	19	
User perception of office space:		
Overcrowded	20	Mean: 2.26 Variance: 0.61 SD: 0.78
Crowded	33	
Not crowded	47	

**Table 2 ijerph-17-01972-t002:** Sick Building Syndrome (SBS) prevalence among staff.

Question	Often	Sometimes	Never	Mean	Standard Deviation
Fatigue	45.98%	42.53%	11.49%	1.66	0.68
Severe headache	19.54%	24.14%	56.32%	2.37	0.79
Headache	27.38%	53.57%	19.05%	1.92	0.68
Nausea/dizziness	7.23%	33.73%	59.04%	2.52	0.63
Difficulty concentrating	27.59%	48.28%	24.14%	1.97	0.72
Itching, burning, irritation of the eyes	15.29%	32.94%	51.76%	2.36	0.74
Irritated, stuffy or runny nose	14.12%	34.12%	51.76%	2.38	0.72
Dry throat	11.90%	44.05%	44.05%	2.32	0.68
Cough	7.14%	48.81%	44.05%	2.37	0.62
Dry or flushed facial skin	11.49%	29.89%	58.62%	2.47	0.70
Scaling/itching scalps/ears	6.10%	17.07%	76.83%	2.71	0.58
Dry hands/itching read skins	9.52%	14.29%	76.19%	2.67	0.65
Others	5.71%	8.57%	85.71%	2.80	0.53

**Table 3 ijerph-17-01972-t003:** User perception of room temperature, noise and lighting.

Question	Often	Sometimes	No	Mean	Standard Deviation	P ^a^
Varying room temperature	41.11%	37.78%	21.11%	1.80	0.77	0.94
Too low room temperature	12.94%	36.47%	50.59%	2.38	0.71	0.39
Stuffy “bad” air	34.78%	33.70%	31.52%	1.97	0.82	0.50
Dry air	15.56%	35.56%	48.89%	2.33	0.73	0.06
Unpleasant smell	18.89%	28.89%	52.22%	2.33	0.78	0.09
Passive smoking	3.41%	12.50%	84.09%	2.81	0.48	0.42
Noise	53.19%	32.98%	13.83%	1.61	0.72	0.55
Dim lighting	8.05%	19.54%	72.41%	2.64	0.63	0.66
Glare/reflection on work surface	25.56%	21.11%	53.33%	2.28	0.85	0.42
Dust and dirt	24.72%	44.94%	30.34%	2.06	0.74	0.77

^a^*p* value in Pearson chi square test.

**Table 4 ijerph-17-01972-t004:** Average measurement of indoor air quality in the selected office environments.

Variables	Acceptable Limit	Office 1	Office 2	Office 3
Temperature (^o^C)	19–22 ℃ *	25.8	22	20.9
Relative humidity (RH%)	40–70%	39.9	47.6	46
CO_2_ (ppm)	<1000 ^‡^	630	448	608
CO (ppm)	30 †	0.0	0.1	0.0
TVOC (ppb)		105	272	126
Mold (CFU/M^3^)		398	833	443

* Based on guidance issued by the Health and Safety Executive, it is reasonable to maintain a temperature around 19 °C; other guidance for sedentary occupations suggests between 19 and 21 °C during winter and 20 to 22 °C in summer as the comfort zone. ^‡^ Acceptable concentrations typical of occupied indoor spaces with good air exchange (OSHA technical manual section iii, chapter 2, 1999). † Long term exposure limit (8-hr TWA reference period). EH40/2005 Workplace exposure limit.

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
