# Peer review of "Assessment of Indoor Air Quality and Users Perception of a Renovated Office Building in Manchester"

_ijerph, 2020, doi:10.3390/ijerph17061972_

Round 1

Reviewer 1 Report

Please make changes to the manuscript as shown on the attached PDF copy.

Author Response

Rebuttal: Reviewer 1

Many thanks for the time taken to review and proof read the work. The input made to our submission is greatly acknowledge by both authors.

Since there was no change requested but pointers made toward grammar and structure, all suggested changes has now been made as evidenced in the revised version submitted. 

Reviewer 2 Report

The topic is interesting and I appreciate the authors work on this topic... but the authors should reorganize and rewrite the structure of the paper.

At first, they speak about a space that is not argued ! It is a very high mistake. The space were users were analyzed MUST BE ANALYZED and DESCRIBED. If it can be useful I suggest to refer to this paper - related to hospital design - in which the rooms analyzed are described and there are the plans, the furniture, the sun exposure..

please refer (this is just an example) to:

  • Gola M, Settimo G, Capolongo S (2019) Indoor air in healing environments: Monitoring chemical pollution in inpatient rooms. Facilities; 37(9/10):600-623. DOI: 10.1108/F-01-2018-0008)

This topic is very important, because the IAQ is highly affected also by the space (its configuration, sun exposure, ...), the furniture and finishing materials, windows opening and/or HVAC, etc. and should in any case highlight this aspect (as suggestions refer to:

  • SmieÅ‚owska M, M. Mar´c, and B. ZabiegaÅ‚a, “Indoor air quality in public utility environments—a review,” Environmental Science and Pollution Research, vol. 24, no. 12, pp. 11166–11176, 2017.
  •  Gola M, Settimo G, Capolongo S (2019) Indoor Air Quality in Inpatient Environments: A Systematic Review on Factors that Influence Chemical Pollution in Inpatient Wards. Journal of Healthcare Engineering; 8358306. Doi: 10.1155/2019/8358306.)

The aim of the paper is clear, but the outcome of the paper cannot be the data analysis of only one case study. From my point of view it should be the definition of a protocol for a wide and broad application by the authors, or by other research groups !Therefore my suggestion is to add the protocol of the questions and all the technical aspects related to the air sampling and monitoring.

Another aspect TO BE CONSIDERED is the sampling activities:

  • how did you assess the data.. just one week, is it enough ?
  • what type of tools you used... do the authors know the sensibility of the instrumentation ? (does the TVOC tool responds to the ISO 16000 ?)
  • where did you localized (on the table, near the window...) >>> the terms locations 1,2,3 are not enough, they need to be described
  • how many people were in the rooms when you did the samplings..

THIS PART IS WRONG and LACKS in the PAPER: sections 2.2 and 2.3 need to be more and more argued, and they should be argued  in the discussion

it is not necessary to repeat regularly the terms SICK BUILDING SINDROME. After the first time you use it, you can use the acronym SBS.

Attention to the terms you used TVOC and VOC: what type of compounds are you investigated ? Attention because they are different ! the authors should use the correct terms, and it is required they know what they are investigating. Please,for supporting the authors, refer to the following work:

  • Settimo, G. Existing guidelines in indoor air quality: the case study of hospital environments. In Indoor air quality in Healthcare Facilities, 1st ed.; Capolongo, S., Settimo, G., Gola, M., Eds.; Springer Public Health: New York City, New York, USA, 2017; pp. 13-26. DOI: 10.1007/978-3-319-49160-8_2

If you analyze only one case study, I suggest you add in the title "in UK" or "in Manchester"

In conclusion, I suggest the authors should have a look to:

  1. Azara, A.; Dettori, M.; Castiglia, P.; Piana, A.; Durando, P.; Parodi, V.; Salis, G.; Saderi, L.; Sotgiu, G. Indoor Radon Exposure in Italian Schools. J. Environ. Res. Public Health 2018, 15, 749.
  2. Settimo G (2012) Residential indoor air quality: significant parameters in light of the new trends. Igiene e sanità pubblica; 68(1):136-138.
  3. Settimo G, D’Alessandro D. European community guidelines and standards in indoor air quality: what proposals for Italy. Epidemiol Prev. 2014;38(6):36–41.

Author Response

Many thanks for the kind comments raised which has helped to improve the paper quality. Find attached our rebuttals to each point raised. 

Round 2

Reviewer 2 Report

I really appreciate the work you have done.

The paper with the recent additional information works well.

My consideration related to the possibility to do a broad and wide application on several case studies.

Only one more integration: a figure with the plan of the 3 spaces that you have analyzed, with the general features (windows, doors, furniture, materials..).

You have not mentioned these information, but the data analysis and data collection are highly affected also by these design aspects. 

Author Response

Dear reviewer, 

Many thanks for your input. Your recommedation has now been provided in section 2.2 "The study area"
